# Immune Responses to COVID-19 Vaccines in Patients with Chronic Kidney Disease and Lead Exposure

**DOI:** 10.3390/ijms232315003

**Published:** 2022-11-30

**Authors:** Ju-Shao Yen, Yao-Cheng Wu, Ju-Ching Yen, I-Kuan Wang, Jen-Fen Fu, Chao-Min Cheng, Tzung-Hai Yen

**Affiliations:** 1Department of Dermatology, Chang Gung Memorial Hospital, Linkou Branch, Taoyuan 333, Taiwan; 2Department of Nephrology, Clinical Poison Center, Chang Gung Memorial Hospital, Linkou Branch, Taoyuan 333, Taiwan; 3School of Medicine, College of Medicine, China Medical University, Taichung 406, Taiwan; 4Department of Nephrology, China Medical University Hospital, Taichung 404, Taiwan; 5Department of Medical Research, Chang Gung Memorial Hospital, Linkou Branch, Taoyuan 333, Taiwan; 6College of Medicine, Chang Gung University, Taoyuan 333, Taiwan; 7Institute of Biomedical Engineering, National Tsing Hua University, Hsinchu 300, Taiwan

**Keywords:** COVID-19, chronic kidney disease, blood lead level, granulocyte-colony stimulating factor, interleukin-8, monocyte chemoattractant protein-1, macrophage inflammatory protein-1α

## Abstract

Literature data regarding the response rate to COVID-19 vaccination in chronic kidney disease (CKD) patients remain inconclusive. Furthermore, studies have reported a relationship between lead exposure and susceptibility to viral infections. This study examined immune responses to COVID-19 vaccines in patients with CKD and lead exposure. Between October and December 2021, 50 lead-exposed CKD patients received two doses of vaccination against COVID-19 at Chang Gung Memorial Hospital. Patients were stratified into two groups based on the median blood lead level (BLL): upper (≥1.30 μg/dL, n = 24) and lower (<1.30 μg/dL, n = 26) 50th percentile. The patients were aged 65.9 ± 11.8 years. CKD stages 1, 2, 3, 4 and 5 accounted for 26.0%, 20.0%, 22.0%, 8.0% and 24.0% of the patients, respectively. Patients in the lower 50th percentile of BLL had a lower proportion of CKD stage 5 than patients in the upper 50th percentile BLL group (*p* = 0.047). The patients in the lower 50th percentile BLL group also received a higher proportion of messenger RNA vaccines and a lower proportion of adenovirus-vectored vaccines than the patients in the upper 50th percentile BLL group (*p* = 0.031). Notably, the neutralizing antibody titers were higher in the lower 50th percentile than in the upper 50th percentile BLL group. Furthermore, the circulating levels of granulocyte-colony stimulating factor, interleukin-8, monocyte chemoattractant protein-1 and macrophage inflammatory protein-1α were higher in the upper 50th percentile than in the lower 50th percentile BLL group. Therefore, it was concluded that lead-exposed CKD patients are characterized by an impaired immune response to COVID-19 vaccination with diminished neutralizing antibodies and augmented inflammatory reactions.

## 1. Introduction

The morbidity and mortality from COVID-19 and its complications have impelled an unprecedented pace in vaccine development. Currently, four types of vaccine are approved for use in Taiwan. The vaccines can be classified into three categories: adenovirus-vectored vaccines (ChAdOx1-S), lipid nanoparticles encapsulating nucleoside-modified messenger RNA vaccines (BNT162b2 and mRNA-1273) and protein subunit vaccines (MVC-COV1901). Trials and ongoing studies have sought to evaluate the efficacy and safety of these vaccines. High vaccine efficacy against symptomatic laboratory-confirmed COVID-19 has been presented, with 70% after the second dose of the ChAdOx1-S vaccine [1], more than 90% after the second dose of the BNT162b2 vaccine [2] and more than 90% after the second dose of the mRNA-1273 vaccine [3].

Nonetheless, these vaccine trials have excluded immunocompromised groups, such as patients with malignancy, organ transplant recipients, patients with rheumatic disorders, and patients undergoing dialysis, resulting in a paucity of data on the efficacy of vaccines in these groups. Nevertheless, according to one meta-analysis [4], seroconversion after one vaccine dose was approximately half as likely in patients with hematological cancers and solid cancers compared with immunocompetent controls, whereas seroconversion was increasingly likely after the second dose. However, organ transplant recipients were 16 times less likely to seroconvert than immunocompetent controls after one dose, and only one-third achieved seroconversion after the second dose. Similarly, data in the literature regarding the immune response rate to COVID-19 vaccination in chronic kidney disease (CKD) patients remain inconclusive [5]. According to our analysis, the published response rates varied from 29.6% to 96.4% [5]. The variable response rates across these clinical trials may be explained by different vaccine types, vaccine doses, criteria for positive response, timings of antibody detection, and races and ethnicities. Moreover, Anand et al. [6] reported that more than one in five dialysis patients demonstrated an attenuated immune response to COVID-19 vaccination.

The motivation for this study was a critical but currently unsatisfactorily answered question in the literature. In addition to causing systemic inflammation in the body, lead exposure has been demonstrated to be related to impaired respiratory function [7,8]. While direct data linking lead exposure and COVID-19 risk or severity are lacking, some studies have reported a relationship between lead exposure and susceptibility to viral infections [9,10]. Theoretically, CKD affects all parts of the immune system. Immune dysregulation in CKD is characterized not only by immunodepression that contributes to a higher risk of infection but also by immunoactivation that causes a greater risk of cardiovascular disease [11]. Altogether, although patients with CKD and lead exposure are more vulnerable to COVID-19 infection than the general population, no studies have been conducted to determine whether lead exposure affects the immune response to COVID-19 vaccination in CKD patients. Therefore, this study attempted to examine the response rates to COVID-19 vaccines in patients with CKD and lead exposure and to analyze the underlying inflammatory reaction.

## 2. Results

The patients were 65.9 ± 11.8 years old, and 64.0% of the patients were male (Table 1). CKD stages 1, 2, 3, 4 and 5 accounted for 26.0%, 20.0%, 22.0%, 8.0% and 24.0% of patients, respectively. None of the stage 5 CKD patients was on a chronic dialysis program. Systemic and kidney diseases were prevalent and included hypertension (80.0%), dyslipidemia (40.0%), diabetes mellitus (40.0%) and gouty arthritis (22.0%). Patients in the lower 50th percentile of BLL had a lower proportion of CKD stage 5 than patients in the upper 50th percentile BLL group (*p* = 0.047). No significant differences were observed in other baseline variables between the groups.

As shown in Table 2, the mean BLL was 1.9 ± 1.9 μg/dL. The patients in the upper 50th percentile of BLL had higher BLLs than the patients in the lower 50th percentile of BLL (3.1 ± 2.3 versus 0.9 ± 0.2, *p* < 0.001). The patients in the upper 50th percentile of BLL had higher creatinine than the patients in the lower 50th percentile of BLL (4.7 ± 4.3 versus 1.9 ± 2.8, *p* = 0.011). No significant differences were found in other laboratory variables between the two groups.

The ChAdOx1-S vaccine, BNT162b2 vaccine, mRNA-1273 vaccine and MVC-COV1901 vaccine accounted for 34.0%, 64.0%, 0%, and 2.0% of patients, respectively (Table 3). The patients in the lower 50th percentile BLL group received a higher proportion of messenger RNA vaccine and a lower proportion of adenovirus-vectored vaccine than the patients in the upper 50th percentile BLL group (*p* = 0.031).

In an analysis of neutralizing antibodies against COVID-19 after vaccination (Figure 1), the neutralizing antibodies were higher in the lower 50th percentile BLL group than in the upper 50th percentile BLL group. A two-way ANOVA was performed to analyze the effect of CKD stage and BLL on COVID-19 neutralizing antibody titers. A simple effects analysis showed that CKD stage did not have a statistically significant effect on COVID-19 neutralizing antibody titers (*p* = 0.523). Nevertheless, the analysis showed that BLL did have a statistically significant effect on COVID-19 neutralizing antibody titers (*p* = 0.017).

In an analysis of inflammatory biomarkers, circulating granulocyte-colony stimulating factor (G-CSF), interleukin (IL)-8, monocyte chemoattractant protein-1 (MCP-1) and macrophage inflammatory protein-1α (MIP-1α) were higher in the upper 50th percentile BLL group than in the lower 50th percentile BLL group (Figure 2).

On the other hand, no significant differences were found in the circulating levels of interferon-γ (IFN-γ), IL-1β, interleukin-1 receptor antagonist (IL-1RA), IFN-γ-inducible protein 10 (IP-10) and tumor necrosis factor-α (TNF-α) between the two groups (Figure 3).

## 3. Discussion

The analytical results revealed that lead-exposed CKD patients are characterized by an impaired immune response to COVID-19 vaccination with diminished neutralizing antibodies and augmented inflammatory reactions. The data are important not only because COVID-19 neutralizing antibodies were higher in the lower 50th percentile BLL group than in the upper 50th percentile BLL group, but also because circulating G-CSF, IL-8, MCP-1 and MIP-1α were higher in the upper 50th percentile BLL group than in the lower 50th percentile BLL group.

The damaging effect of lead on the immune system has been studied. The immune system comprises the innate and adaptive arms. Lead affects neutrophils and macrophages, which are integral to the innate arm of the immune system. A dose-dependent relationship of blood lead on absolute neutrophil count has been observed in occupationally exposed workers [12]. Bone marrow-derived macrophages have been reported to amplify the production of tumor necrosis factor-alpha, interleukin-6, interleukin-12, and prostaglandin E2 and reduce the production of anti-inflammatory interleukin-10 upon exposure to lead [13]. Lead has been shown to decrease phagocytosis and chemotaxis of macrophages and affect nitric oxide production and eicosanoid metabolism in mature macrophages. Blood lead has been shown to increase phagocytosis of erythrocytes and decrease the expression of interferon-gamma-inducible GTPases, p65-GBP, and p47-IRG, which are critical for killing intracellular pathogens [14]. Lead also disturbs the humoral arm of the immune system. Increased lead exposure has been revealed to not only depress total antibody levels but also skew antibody isotype production. Lead exposure has been revealed to result in switching of B lymphocytes from producing immunoglobulin (Ig)M and IgG antibody isotypes to producing IgE isotypes [15,16]. IgA levels have also been found to be increased significantly in individuals with BLLs greater than 10 μg/dL [17]. T lymphocytes appear vulnerable to the pernicious effects of lead as well. Mishra et al. [18] reported a reduction in cluster of differentiation (CD)4 + T cells in lead-exposed patients and found a significant negative correlation between the CD4 + percentage and BLLs. Lead-exposed dendritic cells have been shown to polarize antigen-specific T cells to T helper (Th)2 cells [19]. While short-term exposure to lead has not been demonstrated to affect cytokines related to Th1-, Th2-, and Th17-mediated immune responses, chronic exposure has been revealed to modify their levels [20]. Although COVID-19 neutralizing antibodies are likely to be crucial in vaccine-induced protection, evidence also points to a role for other immune effector mechanisms, including non-neutralizing antibodies, T cells and innate immune mechanisms [21]. Therefore, the present study evaluates the malicious effect of lead exposure in terms of the immune response to COVID-19 vaccines from multiple aspects, including neutralizing antibodies and inflammatory cytokines.

Although the influence of lead on the immune system is well established, there are limited data regarding the potential impact of lead on functional immunity. In 1994, Lutz et al. [22] demonstrated that among children aged 9 months to 6 years recruited through the Women, Infants and Children program in Missouri, BLLs were associated with decreased vaccine-induced antibody titers for diphtheria and rubella but not tetanus. More recent studies of children aged 3 to 7 years living in an electronic waste recycling area in Guangdong Province, China, found that BLLs were associated with lower IgG antibody titers against pertussis, diphtheria, polio, measles [23] and hepatitis B [23,24]. However, these children were exposed to high levels of various other pollutants that could also disturb the immune system. Nonetheless, all of these studies were conducted in populations with moderate to high exposure. Most recently, Di Lenardo et al. [25] revealed that among children participating in the Venda Health Examination of Mothers, Babies and their Environment program in South Africa, low BLLs (1.90 μg/dL) were associated with higher risks of having IgG titers below the protective limit for tetanus but not measles or Haemophilus influenzae type b. The BLLs were also associated with low Haemophilus influenzae type b IgG titers among children exposed to HIV in utero and with low measles IgG titers among females. In the present study, COVID-19 neutralizing antibodies were higher in the lower 50th percentile BLL group than in the upper 50th percentile BLL group, highlighting the contribution of lead exposure to vaccine efficacy.

The ChAdOx1-S and mRNA-1273 vaccines accounted for 98.0% of patients (Table 3). Although they belong to two different categories, both ChAdOx1-S and mRNA-1273 vaccination stimulate a predominantly Th1-type response. A recent study revealed that homologous mRNA-1273 vaccination provoked higher neutralizing antibody titers than homologous ChAdOx1-S vaccination in healthy adults [26]. Interestingly, patients in the lower 50th percentile BLL group received a higher proportion of messenger RNA vaccine and a lower proportion of adenovirus-vectored vaccine than patients in the upper 50th percentile BLL group, which could also lead to the higher neutralizing antibody titers.

Circulating G-CSF, IL-8, MCP-1 and MIP-1α were higher in the upper 50th percentile BLL group. G-CSF is a hematopoietic growth factor that regulates neutrophil production. The clinical use of G-CSF has been explored in numerous disease states, such as to stimulate the production of neutrophils in chemotherapy-related neutropenia and to mobilize hematopoietic stem cells from the bone marrow into the blood to enhance the safety and efficacy of hematopoietic stem cell transplantation [27]. Di Lorenzo et al. [28] demonstrated that circulating G-CSF levels of 33 male lead-exposed workers at a lead recycling plant were higher than those of 28 nonexposed males and that circulating G-CSF levels were correlated with BLLs and absolute neutrophil count. IL-8 is a proinflammatory cytokine and a potent chemotactic agent for neutrophils produced by several cell types. It has been implicated in angiogenesis and metastasis in many in vitro and in vivo models [29]. Lin et al. [30] revealed that divalent lead induced IL-8 gene expression by extracellular signal-regulated kinases and the transcription factor activator protein 1 in human gastric carcinoma cells. Yang et al. [31] reported a positive relationship between lead exposure and increased IL-8 concentration among children living near a lead refinery in Sichuan, China. IL-8 was also significantly higher after subchronic exposure to lead in male workers [32]. MCP-1 is a strong chemotactic factor for monocytes that regulates the migration and infiltration of monocytes, memory T lymphocytes, and natural killer cells. It has been revealed to be involved in various diseases, including multiple sclerosis, rheumatoid arthritis, atherosclerosis, insulin-resistant diabetes, HIV neurological complications and tumor neovascularity [33]. Stimulation of mouse microglia with lead caused upregulation of extracellular signal-regulated kinase and protein kinase B pathways, along with activation of nuclear factor-κB, leading to increased levels of MCP-1 [34]. Exposure to lead acetate in rats has been shown to cause elevated gene expression of MCP-1 [35]. MIP-1α is a member of the MIP-1 CC chemokine subfamily. MIP-1 proteins are known for their chemotactic and proinflammatory effects but can also promote homoeostasis [36]. An in vitro investigation of human peripheral blood mononuclear cells showed that lead exposure resulted in a dose-dependent increase in MIP-1α production by peripheral blood mononuclear cells [37]. A positive correlation between BLL and circulating MIP-1α has been observed in 36 male workers exposed to lead within a short period of time [38].

Circulating concentrations of IFN-γ, IL-1β, IL-1RA, IP-10 and TNF-α were similar between the upper 50th percentile BLL and lower 50th percentile BLL groups. There was no clear explanation. Among them, IFN-γ, IL-1β and TNF-α are Th1-type cytokines. Studies have demonstrated that strong Th1-biased T-cell responses can drive protective humoral and cell-mediated immune responses [39]. Ewer et al. [40] observed an induction of a Th1-biased response characterized by IFN-γ and TNF-α cytokine secretion by CD4+ T cells and antibody production predominantly of IgG1 and IgG3 subclasses up to 8 weeks after vaccination with a single dose of ChAdOx1-S. Similarly, mRNA-1273 was shown to elicit Th1-skewed T-cell responses after the first dose, with 0.05% of circulating CD4+ T cells secreting TNF-α and IL-2 following in vitro stimulation with S protein peptides [41]. However, this Th1-skewed immune response was not observed in the present study. Nevertheless, studies have observed correlations between these cytokines and COVID-19 vaccination. Bergamaschi et al. [42] reported that systemic IL-15, IFN-γ, and IP-10 signatures were associated with effective immune responses to COVID-19 in BNT162b2 vaccine recipients. Tahtinen et al. [43] demonstrated that the IL-1–IL-1RA axis regulated vaccine-mediated systemic inflammation in a host-specific manner and that RNA vaccines induced the production of IL-1 cytokines, which in turn triggered the induction of the broad spectrum of proinflammatory cytokines in human immune cells.

In Taiwan, the phasing out of lead in petrol started in 1983, and the supply of leaded petrol was banned in 2000. Nevertheless, lead persists in the environment as a toxicant [44]. Apart from occupational exposure, the main sources of body lead in the general population include ingestion (drinking water, paints, food and beverages) and inhalation (factory emission and automobile exhausts). In addition, the recruited patients in this study were divided into two groups based on a cutoff BLL of 1.3 μg/dL. Although a blood lead reference value of 10 μg/dL is commonly used in adults, no safe blood lead level has been recognized [45]. Therefore, the harmful effects of lead at any detectible concentration should not be ignored. This study emphasized the concept of no safe BLL and disclosed an impaired immune response to COVID-19 vaccination of lead, even at low BLLs.

The weakened immune response to vaccination could lead to reduce resistance to the development of new forms of SARS CoV-2 infection, such as the Omicron variant that became dominant globally. Omicron, which spread more quickly than any other variant, is a variant of concern. The higher transmissibility can be attributed to its extraordinary power to evade the immunity acquired by both vaccination and previous infections [46]. Nevertheless, our study is limited by not performing an effectiveness study of vaccines against COVID-19 infection, especially the Omicron variant. Moreover, this study is also limited by small sample size and lack of serial laboratory testing.

## 4. Materials and Methods

### 4.1. Inclusion and Exclusion Criteria

Between October 2021 and December 2021, a total of 50 lead-exposed CKD patients received two doses of vaccination against COVID-19 at Chang Gung Memorial Hospital. Patients aged less than 18 years of age, patients with occupational lead exposure, and patients who were hospitalized or underwent surgeries within 3 months were excluded from the analysis.

### 4.2. Study Design

The recruited patients were stratified into two groups based on the median BLL: upper (≥1.30 μg/dL, n = 24) and lower 50th percentile (<1.30 μg/dL, n = 26). Baseline demographic and clinical and laboratory data were recorded for analysis. Blood samples were collected in the hospital for analysis of BLLs, neutralizing antibodies against COVID-19, and blood cytokine concentrations. The rationale for selection of these pro-inflammatory cytokines is based on the fact that, chronic lead exposure could stimulate a cascade of inflammatory events that might impair immune responses to COVID-19 vaccines.

### 4.3. Measurement of Neutralizing Antibodies against COVID-19

The neutralizing antibody titers were determined with MeDiPro SARS-CoV-2 Antibody ELISA (Formosa Biomedical Technology Co., Ltd. Taiwan). The assay detects antibodies against SARS-CoV-2 viral spike protein 1 and the receptor-binding domain and has been approved by the Taiwan Food and Drug Administration. Briefly, whole blood samples were centrifuged at 1500 rpm for 10 min, and the bottom layer of red blood cells was discarded. The tested samples were diluted 1:200 with serum diluent. Serum diluents (as a blank), calibrators (standard) and tested samples were added to the wells and incubated at 37 °C for 30 min. After washing, 100 μL of horseradish peroxidase-conjugated anti-human IgG was added to each well and incubated at 37 °C for 30 min. After washing, 100 μL of tetramethylbenzidine was added to each well and incubated at 37 °C for 10 min. Then, 100 μL of 1 N hydrochloric acid was added to stop the chemical reaction. Finally, the absorbance was measured at 450 nm against a reference wavelength at 650 nm using a microplate reader. The limit of detection was 12 IU/mL.

### 4.4. Blood Lead Determination Using Graphite Furnace Atomic Absorption Spectrometry

Whole-blood samples were collected in metal-free tubes (Vacutainer; BD 368381; Becton-Dickson, Franklin Lakes, NJ, USA). The BLLs were examined by using graphite furnace atomic absorption spectrometry (PinAAcle 900T; PerkinElmer, Waltham, MA, USA) [44]. Whole-blood samples (200 μL) were diluted (1:4) with a matrix modifier solution. All standards were acquired from High-Purity Standards and traceable to the National Institute of Standards and Technology (Charleston, SC, USA). Calibration was achieved using a reagent blank with five calibration standards. Calibration curves for lead had an R ≥ 0.995 and a blank absorbance < 0.005. The recovery rate for lead was within 90%–110%. The limit of quantification was 1.0 µg/dL. Quality controls were analyzed at the start and end of each analytical run and once per level again after every 10 samples. Test precision was supervised using a coefficient of variation of less than 5% at each level of control. The test accuracy was validated regularly by the College of American Pathologists proficiency testing.

### 4.5. Cytokine Measurements Using Multiplex Immunoassay

Blood samples were obtained, centrifuged, and stored at −80 °C [47]. Blood cytokine concentrations, namely, G-CSF, IFN-γ, IL-1β, IL-1RA, IL-8, IP-10, MCP-1, MIP-1α and TNF-α, were evaluated with a Bio-Plex Human Cytokine Assay Kit (Bio–Rad Laboratories, Hercules, CA, USA). Briefly, 50 µL antibody-coupled beads per well were added to flat-bottom plates and washed twice. Next, 50-µL serum/plasma samples were incubated with antibody-coupled beads for 30 min at room temperature. After washing three times to remove unbound substances, the beads were incubated with 25 µL biotinylated detection antibodies for 30 min at room temperature. After washing the unbound biotinylated antibodies three times, the beads were incubated with 50 µL streptavidin-phycoerythrin for 10 min at room temperature. After cleaning of streptavidin-phycoerythrin by washing three times, the beads were resuspended in 125 µL assay buffer. Beads were read on the Bio-Plex suspension array system, and the results were analyzed with Bio-Plex Manager software version 6.0. The limits of detection were as follows: G-CSF 6.35 pg/mL, IFN-γ 1.57 pg/mL, IL-1β 0.29 pg/mL, IL-1RA 6.21 pg/mL, IL-8 0.85 pg/mL, IP-10 3.41 pg/mL, MCP-1 0.53 pg/mL, MIP-1β 1.41 pg/mL, and TNF-α 3.33 pg/mL.

### 4.6. Statistical Analysis

Continuous variables are reported as the mean with standard deviation. Categorical variables are reported as numbers with percentages in parentheses. All the data were tested for normality of distribution and equality of standard deviations before analysis. The quantile–quantile plot and Kolmogorov–Smirnov test were used to check the normality of distribution. The Levene test was used to check the equality of variance. Comparisons between groups were performed using Student’s t-test for quantitative variables and the chi-squared test for qualitative variables. A two-way analysis of variance (ANOVA) was performed to analyze the effect of CKD stage and BLL on COVID-19 neutralizing antibody titers. A *p* value of less than 0.05 was considered statistically significant. All analyses were performed with IBM SPSS Statistics Version 20.0.

## 5. Conclusions

In conclusion, our analysis found that lead exposure is characterized by an impaired immune response to COVID-19 vaccination with diminished neutralizing antibodies and increased systemic inflammation that involves elevated levels of circulating G-CSF, IL-8, MCP-1 and MIP-1α.

## Figures and Tables

**Figure 1 ijms-23-15003-f001:**
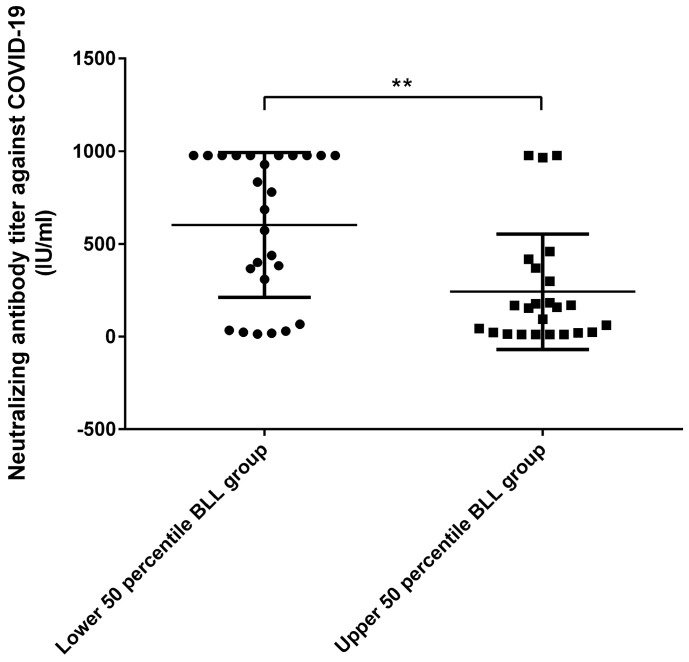
Analysis of neutralizing antibody titers against COVID−19. The neutralizing antibody titer was lower in the upper 50th percentile blood lead level (BLL) group than in the lower 50th percentile BLL group. ** *p* < 0.01.

**Figure 2 ijms-23-15003-f002:**
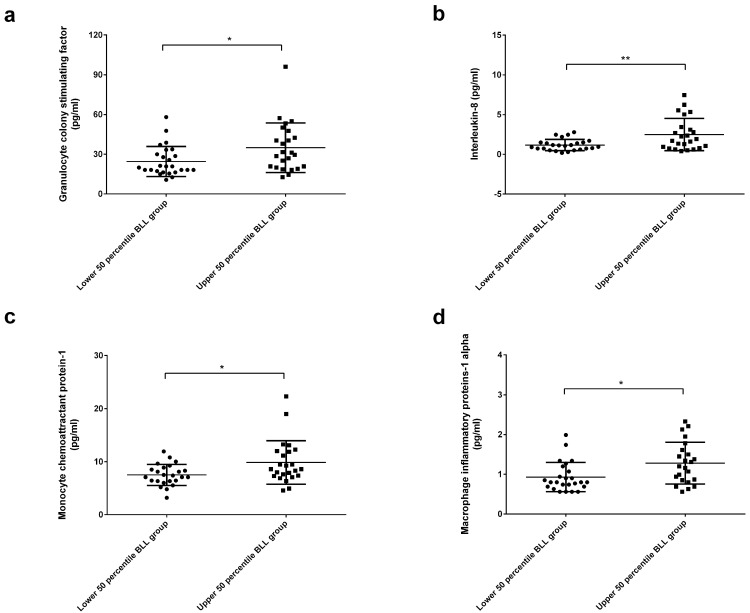
Analysis of inflammatory biomarkers. (**a**) Circulating granulocyte-colony stimulating factor (G−CSF), (**b**) Interleukin (IL)−8, (**c**) Monocyte chemoattractant protein−1 (MCP−1), (**d**) Macrophage inflammatory protein−1α (MIP−1α)). The G−CSF, IL−8, MCP−1 and MIP−1α were higher in the upper 50th percentile BLL group than in the lower 50th percentile BLL group. * *p* < 0.05, ** *p* < 0.01.

**Figure 3 ijms-23-15003-f003:**
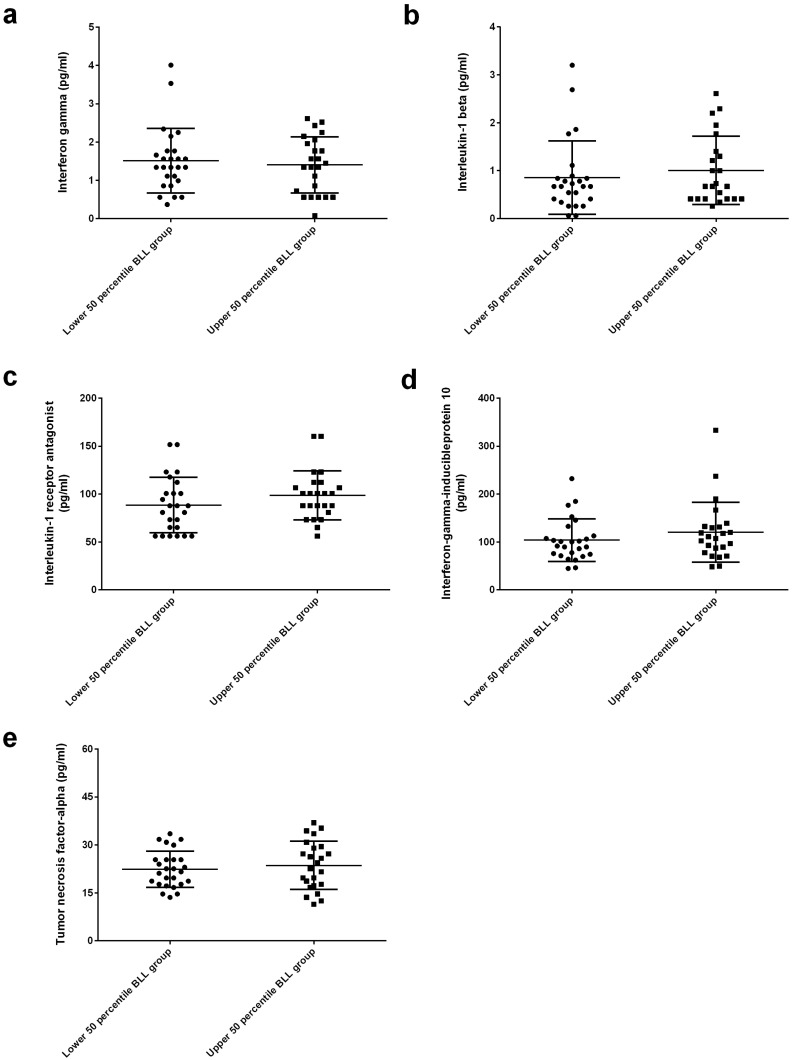
Analysis of inflammatory biomarkers. (**a**) Interferon−γ (IFN−γ), (**b**) Interleukin 1β (IL−1β), (**c**) Interleukin−1 receptor antagonist (IL−1RA), (**d**) Interferon-γ-inducible protein 10 (IP−10), (**e**) Tumor necrosis factor−α (TNF−α). There were no significant differences in circulating levels of IFN−γ, IL−1β, IL−1RA, IP−10 and TNF−α between patients in the upper 50th percentile and lower 50th percentile blood lead level (BLL) groups.

**Table 1 ijms-23-15003-t001:** Baseline demographics of patients stratified by median blood lead level (BLL) as the upper or lower 50th percentile (n = 50).

Variable	Total (n = 50)	Patients with Upper 50th Percentile BLL (n = 24)	Patients with Lower 50th Percentile BLL (n = 26)	*p* Value
Demographics				
Male, n (%)	32 (64.0)	18 (75.0)	14 (53.8)	0.119
Age, year	65.9 ± 11.8	68.1 ± 9.1	63.8 ± 12.8	0.180
Body mass index, kg/m^2^	26.1 ± 4.4	26.2 ± 3.3	26.0 ± 5.3	0.199
Smoking habit, n (%)	10 (20.0)	7 (29.2)	3 (11.5)	0.119
Alcohol consumption, n (%)	8 (16.0)	5 (20.8)	3 (11.5)	0.370
Betel nut usage, n (%)	1 (2.0)	1 (4.2)	0 (0)	0.293
Chronic kidney disease staging				
Stage 1, n (%)	13 (26.0)	3 (12.5)	10 (38.5)	0.054
Stage 2, n (%)	10 (20.0)	5 (20.8)	5 (19.2)	1.000
Stage 3, n (%)	11 (22.0)	5 (20.8)	6 (23.1)	1.000
Stage 4, n (%)	4 (8.0)	2 (8.3)	2 (7.7)	1.000
Stage 5, n (%)	12 (24.0)	9 (37.5)	3 (11.5)	0.047 *
Diabetic mellitus, n (%)	20 (40.0)	12 (50.0)	8 (30.8)	0.166
Hypertension, n (%)	40 (80.0)	21 (87.5)	19 (73.1)	0.203
Biopsy-proved chronic glomerulonephritis, n (%)	4 (8.0)	1 (4.2)	3 (11.5)	0.337
Biopsy-proved tubulointerstitial nephritis	0 (0)	0 (0)	0 (0)	1.000
Polycystic kidney disease, n (%)	0 (0)	0 (0)	0 (0)	1.000
Gouty arthritis, n (%)	11 (22.0)	6 (25.0)	5 (19.2)	0.623
Dyslipidemia, n (%)	20 (40.0)	7 (29.2)	13 (50.0)	0.133
Malignancy, n (%)	3 (6.0)	1 (4.2)	2 (7.7)	0.600
Solid organ transplant, n (%)	0 (0)	0 (0)	0 (0)	1.000
Immunosuppressive therapy, n (%)	0 (0)	0 (0)	0 (0)	1.000

Note: * *p* < 0.05.

**Table 2 ijms-23-15003-t002:** Laboratory data of patients stratified by median blood lead level (BLL) as the upper or lower 50th percentile (n = 50).

Variable	Total (n = 50)	Patients with Upper 50th Percentile BLL (n = 24)	Patients with Lower 50th Percentile BLL (n = 26)	*p* Value
Blood lead level, μg/dL	1.9 ± 1.9	3.1 ± 2.3	0.9 ± 0.2	<0.001 ***
Hemogram				
Hemoglobin, g/dL	11.8 ± 2.7	11.5 ± 2.9	12.0 ± 2.6	0.582
Hematocrit, %	36.0 ± 8.2	35.6 ± 9.2	36.4 ± 7.3	0.776
Red blood cell count, 10^6^/μL	4.2 ± 0.8	4.1 ± 1.1	4.2 ± 0.7	0.822
Mean corpuscular volume, fL	90.0 ± 7.2	91.6 ± 5.2	89.4 ± 8.0	0.577
Platelet count, 10^3^/μL	216.7 ± 68.5	197.6 ± 38.2	224.0 ± 77.2	0.481
White blood cell count, 10^3^/μL	8.6 ± 8.4	6.8 ± 1.6	9.2 ± 9.9	0.602
Biochemistry				
Blood urea nitrogen, mg/dL	32.6 ± 23.3	39.2 ± 23.3	27.8 ± 22.6	0.115
Creatinine, mg/dL	3.2 ± 3.8	4.7 ± 4.3	1.9 ± 2.8	0.011 *
Albumin, g/dL	3.9 ± 0.3	3.9 ± 0.3	3.9 ± 0.3	0.826
Uric acid, mg/dL	6.6 ± 2.0	6.6 ± 1.9	6.5 ± 2.1	0.838
Alanine aminotransferase, U/L	22.3 ± 11.1	19.8 ± 7.8	24.2 ± 12.9	0.252
Total cholesterol, mg/dL	200.3 ± 38.7	183.5 ± 31.2	210.0 ± 40.0	0.069
High-density lipoprotein cholesterol, mg/dL	53.7 ± 15.3	50.7 ± 12.7	55.4 ± 16.7	0.432
Low-density lipoprotein cholesterol, mg/dL	115.8 ± 34.3	111.0 ± 27.2	119.3 ± 39.0	0.502
Triglyceride, mg/dL	142.7 ± 91.8	119.8 ± 44.2	158.3 ± 112.2	0.252
Fasting blood sugar, mg/dL	115.6 ± 33.8	119.5 ± 30.4	112.9 ± 36.6	0.567
Glycated hemoglobin, %	6.3 ± 0.9	6.1 ± 0.9	6.7 ± 0.8	0.259

Note: * *p* < 0.05, *** *p* < 0.001.

**Table 3 ijms-23-15003-t003:** COVID-19 vaccination information of patients stratified by median blood lead level (BLL) as the upper or lower 50th percentile (n = 50).

Variable	Total (n = 50)	Patients with Upper 50th Percentile BLL (n = 24)	Patients with Lower 50th Percentile BLL (n = 26)	*p* Value
Time elapsed between second vaccination and blood test (day)	26.1 (16.4)	30.8 ± 12.6	21.8 ± 18.5	0.051
Type of vaccine				0.031 *
ChAdOx1-S, n (%)	17 (34.0)	12 (50.0)	5 (19.2)	
mRNA-1273, n (%)	32 (64.0)	11 (45.8)	21 (80.8)	
BNT162b2, n (%)	0 (0)	0 (0)	0 (0)	
MVC-COV1901, n (%)	1 (2.0)	1 (4.2)	0 (0)	

Note: Currently, there are four types of vaccine approved for use in Taiwan. The vaccines can be classified into three categories: adenovirus-vectored vaccines (ChAdOx1-S), lipid nanoparticles encapsulating nucleoside-modified messenger RNA vaccines (BNT162b2 and mRNA-1273) and protein subunit vaccines (MVC-COV1901). * *p* < 0.05.

## Data Availability

The datasets used and analyzed in this study are available from the corresponding author upon request.

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
