# Peer review of "Immune Responses to COVID-19 Vaccines in Patients with Chronic Kidney Disease and Lead Exposure"

_ijms, 2022, doi:10.3390/ijms232315003_

Round 1
Reviewer 1 Report
This is a manuscript by Yen et al. evaluating humoral response of chronic kidney disease patients to vaccination against SARS-CoV-2, based on lead levels. Although the concept of the paper is very attractive, I believe that there are several issues that have to be addressed by the authors.
Major comments
1) It is described that there is a significant difference in categories of received vaccine between the two patients categories (table 3). Please comment on potential differences in immunogenicity of different vaccines categories, as this might be a reason of response variation between the two patients cohort.
2) Please describe the plan of the study briefly in the methods sections, justifying the choice of the measured cytokines.
3) It is reported that 4 patients with chronic glomerulonephritis were included in the study. Administration of immunosuppressive treatment might alter humoral response. Please comment on this.
4) Please perform chi square tests for data on CDK stage in table 1. The test should be performed separately for each stage, for example in one crosstab: stage 1 for upper BBL, stage 1 for lower BBL, stages 2-4 for upper BBL, stages 2-4 for lower BBL, and so on for the rest stages. This is important because the cohort with upper BBL appears to have worse kidney function, which is related to both higher lead levels and worse response to vaccination. Moreover, a multivariate anova could be performed to show weather CKD stage interferes with lead levels to response level.
Minor comments
1) It is stated that lead source was not occupational exposure. Please describe briefly any other potential sources, if any known.
2) Please explain 1+4 in line 276. Is it a dilution ratio?
3) Please provide all units in lines 300-301
4) in the methods section it is stated that the authors used T test for comparison. Are the data normally distributed? Please comment respectively.
5) Finally, you should describe stage 5 CKD patients. Are they on dialysis or not? What protocol was used for blood sampling? (i.e. in the initiation of a dialysis session)
Author Response
This is a manuscript by Yen et al. evaluating humoral response of chronic kidney disease patients to vaccination against SARS-CoV-2, based on lead levels. Although the concept of the paper is very attractive, I believe that there are several issues that have to be addressed by the authors.
Major comments
1) It is described that there is a significant difference in categories of received vaccine between the two patients categories (table 3). Please comment on potential differences in immunogenicity of different vaccines categories, as this might be a reason of response variation between the two patients cohort.
Response: Thank you for the comment. The important issue has been addressed in the Discussion paragraph.
The ChAdOx1-S and mRNA-1273 vaccines accounted for 98.0% of patients (Table 3). Although they belong to two different categories, both ChAdOx1-S and mRNA-1273 vaccination stimulates a predominantly Th1-type response. A recent study revealed that homologous mRNA-1273 vaccination provoked higher neutralizing antibody titers than homologous ChAdOx1-S vaccination in healthy adults. Interestingly, patients in the lower 50th percentile BLL group received a higher proportion of messenger RNA vaccine and a lower proportion of adenovirus-vectored vaccine than patients in the upper 50th percentile BLL group, which could also lead to the higher neutralizing antibody titers.
2) Please describe the plan of the study briefly in the methods sections, justifying the choice of the measured cytokines.
Response: Thank you for the comment. The information has been added.
Baseline demographic, clinical and laboratory data were recorded for analysis. Blood samples were collected in the hospital for analysis of BLLs, neutralizing antibodies against COVID-19 and blood cytokine concentrations. The rationale for selection of these pro-inflammatory cytokines is based on the fact that, chronic lead exposure could stimulate a cascade of inflammatory events that might impair immune responses to COVID-19 vaccines.
3) It is reported that 4 patients with chronic glomerulonephritis were included in the study. Administration of immunosuppressive treatment might alter humoral response. Please comment on this.
Response: Thank you for the comment. None of the patients with chronic glomerulonephritis were under immunosuppressive medications (Table 1).
4) Please perform chi square tests for data on CDK stage in table 1. The test should be performed separately for each stage, for example in one crosstab: stage 1 for upper BBL, stage 1 for lower BBL, stages 2-4 for upper BBL, stages 2-4 for lower BBL, and so on for the rest stages. This is important because the cohort with upper BBL appears to have worse kidney function, which is related to both higher lead levels and worse response to vaccination. Moreover, a multivariate anova could be performed to show weather CKD stage interferes with lead levels to response level.
Response: Thank you for the comment.
As instructed, a Chi-Square test has been performed separately for each stage.Patients in the lower50thpercentile of BLL had lowerproportion of CKDstage 5 than patients in the upper 50thpercentile BLL group (P = 0.047).Furthermore, a two-way ANOVA had been performed to analyze the effect of CKD stageand BLL on COVID-19 neutralizing antibody titers. A simple effects analysis showed that CKD stagedid not have a statistically significant effect on COVID-19 neutralizing antibody titers (P = 0.523). Nevertheless, the analysis showed that BLL did have a statistically significant effect on COVID-19 neutralizing antibody titers (P = 0.017).
Minor comments
1) It is stated that lead source was not occupational exposure. Please describe briefly any other potential sources, if any known.
Response: Thank you for the comment. In Taiwan, the phasing out of lead in petrol started in 1983, and the supply of leaded petrol was banned in 2000. Nevertheless, lead persists in the environment as a toxicant. Apart from occupational exposure, the mainsources of body lead in general population include ingestion (drinking water, paints, food and beverages) and inhalation (factory emission and automobile exhausts).
2) Please explain 1+4 in line 276. Is it a dilution ratio?
Response: Thank you for reminding us. The typo error has been corrected.
3) Please provide all units in lines 300-301
Response: Thank you for reminding us. The units have been added.
4) in the methods section it is stated that the authors used T test for comparison. Are the data normally distributed? Please comment respectively.
Response: Thank you for the comment. The data were normally distributed.
All the data were tested for normality of distribution and equality of standard deviations before analysis. The Quantile-quantile plot and Kolmogorov-Smirnov test were used to check the normality of distribution. The Levene test was used to check the equality of variance.
5) Finally, you should describe stage 5 CKD patients. Are they on dialysis or not? What protocol was used for blood sampling? (i.e. in the initiation of a dialysis session)
Response: Thank you for the comment. None of the stage 5 CKD patients was on chronic dialysis program.

Reviewer 2 Report
1: It is necessary to show in table 1 the importance of other diseases that lead to CKD, primarily hypertension and tubulointerstitial diseases due to the action of lead and these two groups of diseases.
2: It would be significant to show whether the weakened immune response to vaccination also led to reduced resistance to the development of new forms of SARS CoV-2 infection, such as the omicron subvariant that became dominant after your research
Author Response
1: It is necessary to show in table 1 the importance of other diseases that lead to CKD, primarily hypertension and tubulointerstitial diseases due to the action of lead and these two groups of diseases.
Response: Thank you for the comment. The data has been added in Table 1.
2: It would be significant to show whether the weakened immune response to vaccination also led to reduced resistance to the development of new forms of SARS CoV-2 infection, such as the omicron subvariant that became dominant after your research
Response: Thank you for the comment. This important point has been addressed in the Discussion paragraph.
The weakened immune response to vaccination could lead to reduce resistance to the development of new forms of SARS CoV-2 infection, such as the omicron variant that became dominant globally. Omicron is a variant of concern, which spread quicker than any other variant. The higher transmissibility can be attributed to its extraordinary power to evade the immunity acquired by both vaccination and previous infections. Nevertheless, our study is limited by not performing effectiveness study of vaccines against COVID-19 infection, especially the Omicron variant. Moreover, this study is also limited by small sample size and lack of serial laboratory testing.

Round 2
Reviewer 1 Report
Comments are adequately responsed, appropriate for publication